# Exercise-Mediated Skeletal Muscle-Derived IL-6 Regulates Bone Metabolism: A New Perspective on Muscle–Bone Crosstalk

**DOI:** 10.3390/biom15060893

**Published:** 2025-06-18

**Authors:** Chenyu Zhu, Xiaoqing Ding, Min Chen, Jie Feng, Jun Zou, Lingli Zhang

**Affiliations:** 1School of Exercise and Health, Shanghai University of Sport, Shanghai 200438, China; 2311517002@sus.edu.cn (C.Z.); 2321517006@sus.edu.cn (J.F.); junzou@sus.edu.cn (J.Z.); 2School of Athletic Performance, Shanghai University of Sport, Shanghai 200438, China; 2311811002@sus.edu.cn; 3Encephalopathy Department, Shanghai Hospital of Traditional Chinese Medicine, Shanghai 200071, China; chenmin8530@126.com

**Keywords:** IL-6, exercise, bone metabolism, muscle, skeletal muscle–bone crosstalk

## Abstract

Skeletal muscles and bones maintain musculoskeletal system function through their collaborative interaction, whereby muscles regulate bone metabolism via mechanical coupling. An increasing number of studies have shown that various cytokines secreted by skeletal muscles during exercise closely regulate the balance of bone homeostasis. Interleukin-6 (IL-6), one of the first muscle-secreted factors to be discovered, not only plays an important role in regulating the function of the muscle itself but also regulates bone metabolic processes in a bidirectional manner through multiple complex signal transduction pathways, thereby affecting the balance between bone formation and bone resorption. The exact mechanism by which IL-6 regulates bone metabolism is not fully understood, and there are few summaries on how exercise affects bone metabolism through IL-6 from skeletal muscles. Accordingly, this study will take skeletal muscle-derived IL-6 as an entry point to explore how the cross-organ regulatory activities of the muscles targeting bones during exercise affect bone metabolic processes. This study also aims to improve the mechanism of muscle–bone crosstalk under the effect of exercise and provide a theoretical basis and clinical diagnosis and treatment ideas from multiple perspectives for exercise to improve bone health.

## 1. Introduction

Frequent engagement with electronic equipment, sedentary lifestyles, and high-fat, high-sugar diets, driven by the continuous advancement of urbanization and industrialization around the world, pose a significant threat to public health [1]. The concept of “exercise is medicine” was officially proposed by the American College of Sports Medicine in 2007 and has been recognized worldwide [2]. Although regular exercise improves health and prevents chronic diseases, the exact mechanisms remain unclear. Metabolites, such as cytokines, mediators, peptides, and RNAs secreted by organs and tissues during exercise, may help to explain the health-promoting effects of physical activity [3].

When exploring the effects of exercise on the secretion of multiple organ systems in the body, skeletal muscle has been identified as an endocrine organ capable of secreting various cytokines, including the interleukin (IL) family and other peptides with paracrine or endocrine effects, thereby regulating other systems, such as the skeletal system, liver, and fat [4]. The skeletal muscle and bone are integrated organs, primarily coupled through mechanical coupling, in which the bone serves as an attachment point for the muscle, while mechanical loading imposed by muscle contraction simultaneously modulates bone remodeling and metabolic activity. However, the skeletal muscle can also non-mechanically affect bone homeostasis [5]. Bioactive factors secreted by muscles are secreted in the form of endocrine or paracrine and are released into the bone tissue through synaptic vesicles, thereby affecting the metabolism of various cells in the bone tissue [6]. Studies have demonstrated that physical exercise potently induces the skeletal muscle to release a spectrum of bioactive cytokines. IL-6 was the first muscle factor to be discovered; it is released into the blood circulation after exercise when muscle fibers contract, thereby regulating a series of metabolic responses in the body [7]. IL-6 is a multifunctional factor that has a complex role in regulating bone metabolism. Currently, few reports have been published on the involvement of the exercise-mediated muscle factor IL-6 in the regulation of bone metabolism. Therefore, this study will explore the effect of exercise-mediated muscle–bone “crosstalk” on bone metabolism from the perspective of the muscle factor IL-6, providing a theoretical basis for exercise to improve bone health.

## 2. Effects of IL-6 on Bone Homeostasis

### 2.1. IL-6 Structure and Function

IL-6 exhibits multifunctional properties as a cytokine [8]. Studies have shown that the amino acid sequences of other factors, such as hepatocyte growth factor 2 (HGF 2), HGF 3, interferon beta 24, and 26 kDa protein 5 identified from fibroblasts, are the same as IL-6, which confirms the pleiotropic activity of IL-6 [9]. Moreover, IL-6 is considered one of the main immunomodulatory cytokines controlling health and disease [10]. IL-6 is a single-chain phosphorylated glycoprotein composed mainly of four helical bundles (A-D). Furthermore, IL-6 transmits its signals via a cell-surface type I receptor complex consisting of a ligand-binding glycoprotein called IL-6 receptor (IL-6R) and the signal transducer component gp130 [11]. IL-6R exists in two forms, membrane-bound (MIL-6R) and soluble (SIL-6R). MIL-6R is predominantly expressed in hepatocytes, neutrophils, monocytes, etc., and performs normal physiological functions mediated by IL-6 [12]. In addition, MIL-6R is cleaved by the proteolytic action of ADAMTS10 and ADAMTS17, which are members of the metalloproteinase gene family, to produce SIL-6R [13]. IL-6-mediated signaling includes three modes: classic IL-6 signaling, IL-6 trans signaling, and IL-6 cluster signaling. In classic signal transduction, MIL-6R binds to IL-6 to induce the signaling homodimer of the gp130 receptor chain. This type of signaling promotes the anti-inflammatory response of leukocytes and hepatocytes and maintains homeostasis [14,15]. By contrast, in IL-6 trans-signaling, IL-6R binds to IL-6 with lower affinity, forming IL-6–IL-6R dimer complexes that spread throughout the body. Thereafter, the dimer binds to gp130 to form a heterotrimer, which subsequently binds with another heterotrimer to form a hexameric complex that triggers an inflammatory response. IL-6 trans-signaling mediated by SIL-6R has become a classic pathological pattern among the pathological mechanisms of various diseases [16,17]. Finally, in the IL-6 cluster signaling, the IL-6–MIL-6R complex formed on the transmitter cell activates the gp130 subunit on the adjacent receiver cell and may play a role in the activation of pathogenic helper T cells 17 (TH17) [18,19].

### 2.2. IL-6 Regulates Bone Homeostasis

Bone homeostasis is essential for forming and maintaining bone form and repairing damaged bone [20]. Moreover, the equilibrium of bone remodeling hinges on the dynamic interplay between osteoblast-driven bone formation and osteoclast-dependent bone resorption. Osteoblasts mainly differentiate from bone marrow mesenchymal stem cells (BMSCs) and play an osteogenic role in the regulation of bone matrix synthesis, secretion, and mineralization [21]. During the terminal phase of osteogenesis, osteoblasts become embedded within the mineralized bone matrix and undergo terminal differentiation into osteocytes, key regulators of skeletal homeostasis [22]. Osteoclasts are the only bone-resorption cells in the body. These osteoclasts are tissue-specific multinucleated macrophages that differentiate from monocytes or macrophage precursors on or near the surface of the bone [22]. During this process, dysfunction of any cell type can disrupt bone homeostasis, potentially leading to bone-related diseases [23].

IL-6 is produced by various cell types in the bone microenvironment, including osteoblasts, osteoclasts, macrophages, and T cells [12,24,25]. Moreover, IL-6 is the first cytokine found to mediate the bone loss associated with estrogen depletion in mice [26]. Relevant studies in recent years have found that IL-6 plays a bidirectional regulatory role in the homeostatic interplay between osteoblast-driven bone formation and osteoclast-dependent bone resorption (Figure 1).

#### 2.2.1. IL-6 Regulates Bone Resorption

Previous studies have demonstrated that IL-6 can not only promote the production of osteoclasts in rat and human bone marrow cell culture systems but also promote the differentiation and maturation of osteoclasts in osteoblast–osteoclast co-culture systems. However, IL-6 does not significantly affect the differentiation of purified osteoclast precursor cells [27,28,29,30]. The expression of IL-6R on the surface of osteoclast precursor cells is much higher than that on osteoblasts. However, IL-6R on the surface of osteoclast precursor cells does not play a role in IL-6-induced osteoclast differentiation. Only IL-6R on the surface of osteoblasts is necessary for IL-6 to promote osteoclast differentiation [29]. Chowdhury et al. found that in the presence of IL-6, the number of mature osteoclasts was significantly reduced when IL-6R-specific knockout osteoblasts were co-cultured with wild-type primary macrophages. Meanwhile, osteoclast differentiation was enhanced after IL-6R was overexpressed in osteoblasts [31].

Osteocytes are the main source of RANKL in the bone microenvironment compared to osteoblasts [32]. IL-6/sIL-6R signaling is found to promote Janus kinase (JAK) 2 phosphorylation-induced RNAKL secretion in osteoblast-like MLO-Y4 cells and stimulate osteoclast differentiation in a co-culture system of MLO-Y4 cells and RAW264.7 cells [33]. In the case of zoledronic acid treatment, IL-6 can activate the signal transducer and transcription factor of the STAT3 pathway, enhance RANKL expression, and promote the differentiation of bone marrow macrophages to osteoclasts in a co-culture system [34]. Therefore, IL-6 indirectly promotes osteoclast differentiation by inducing osteoblasts, BMSCs, or osteocytes to secrete RANKL, rather than directly affecting osteoclasts and their precursor cells [28].

IL-6, as a multifunctional inflammatory factor, can promote the secretion of other inflammatory factors such as IL-1 and TNF-α by osteoblasts or BMSCs to directly or indirectly contribute to the regulation of osteoclastogenesis [35]. However, *IL-6* knockout mice did not show defects in osteoclast formation or bone mass, suggesting that IL-6 has no a unique destructive effect on normal bone physiology [36]. However, in pathological conditions, IL-6 plays an important role in the production of osteoclasts, as *IL-6*-deficient mice show a slowed loss of bone mass after ovariectomy and an inhibition of osteoclast differentiation in inflammatory arthritis [36]. Therefore, these data strongly suggest that IL-6 has been identified as a significant mediator of bone loss under pathological conditions and a putative regulator of osteoclast-mediated bone resorption. In certain pathological states of osteoblast-like cells, such as those derived from myeloma cells, IL-6R is expressed on the surface, and IL-6 can directly bind to it, inducing the bone resorption activity of these osteoblast-like cells [37]. In addition, IL-6 can induce the differentiation of CD14+ monocytes into osteoclasts only in the presence of macrophage colony-stimulating factor (M-CSF). This effect can be inhibited by anti-gp130 antibodies but not by osteoprotegerin (OPG), indicating that IL-6 can directly induce osteoclast differentiation through a RANKL-independent mechanism [38].

Although previous evidence suggests that IL-6 is insufficient to induce the differentiation of purified osteoclast precursor cells, the addition of IL-6R can regulate the effect of IL-6 on the maturation of osteoclast precursor cells. Kudo et al. treated human peripheral blood mononuclear cells with IL-6 or IL-6R alone. They found that IL-6 alone could not induce differentiation into osteoclasts, while the combination of IL-6 and IL-6R could induce TRAP-positive multinucleated osteoclasts and the formation of bone resorption pits [38]. Further studies found that IL-6, regardless of the presence of sIL-6R, attenuates high-level RANKL-induced osteoclast formation and bone resorption activity. By contrast, the combination of IL-6 and sIL-6R enhances rather than inhibits low-level RANKL-induced osteoclastogenesis. This notion indicates that IL-6 plays a bidirectional regulatory role in RANKL-induced osteoclastogenesis in the presence of sIL-6R. The phosphorylation of NF-κB p65, ERK, and JNK pathways has been found to be selectively potentiated by the IL-6-sIL-6R complex under conditions of low RANKL availability, whereas this complex suppresses the same pathways when RANKL levels are high, leading to regulated osteoclast maturation. The RANKL-dependent regulation of osteoclast maturation by IL-6 trans-signaling may reflect pathological processes in the local microenvironment associated with autoimmune diseases related to bone destruction, such as rheumatoid arthritis. The dichotomy between the stimulatory and inhibitory effects of IL-6 on osteoclastogenesis appears to be context-dependent, particularly influenced by disease stage. During the early phase of the disease, when local RANKL concentrations are comparatively low, the elevated secretion of IL-6 and its sIL-6R by activated immune cells enhances osteoclast differentiation and function through dual mechanisms: (1) the indirect stimulation of stromal cells or osteoblasts to upregulate RANKL production, and (2) the direct facilitation of osteoclast precursor commitment toward mature osteoclasts. These processes collectively amplify osteoclast-mediated bone resorption, ultimately contributing to bone and cartilage degradation. Conversely, in advanced- or late-stage disease, when RANKL levels are considerably elevated, IL-6 and sIL-6R may serve a counterregulatory role by mitigating excessive RANKL-driven osteoclast maturation. This dynamic interplay suggests that the IL-6-sIL-6R axis acts as a homeostatic modulator, potentially limiting pathological bone resorption. Thus, the IL-6-sIL-6R complex may represent a protective mechanism against excessive bone loss under the conditions of high RANKL availability [39]. Although IL-6 alone does not affect low-level RANKL-induced osteoclast differentiation, the binding of IL6 and sIL-6R enhances it. This unique effect may be due to the complementary effect of sIL-6R, which replaces the binding of IL-6R to IL-6 and forms a complex with gp130 to initiate IL-6-induced receptor signal transduction, thus enhancing the IL-6 trans-signaling system to a level that significantly affects the differentiation of RANKL osteoclast precursors to mature osteoclasts [40].

In summary, the specific mechanism by which IL-6 regulates osteoclast differentiation, in physiological and pathological conditions, remains to be further explored. Clarifying the role of IL-6 in osteoclast maturation will help us deepen our understanding of the pathological mechanisms of inflammatory bone resorption, thereby providing new targets for bone metabolic diseases caused by osteoclast dysfunction.

#### 2.2.2. IL-6 Regulates Bone Formation

IL-6 affects the differentiation of osteoblasts and bone formation through a complex mechanism, playing a dual role in osteoblast activity. Studies have found that IL-6 transgenic mice have significantly reduced bone mass, severe degeneration of the structure of the trabecular bone and cortical bone, and severely impaired development of the growth plate and epiphyseal ossification center, accompanied by a decrease in the number of osteoblasts and an increase in the activity of osteoclasts [41]. Furthermore, IL-6 expression was significantly increased in the femurs of mice with osteoporosis caused by tail suspension, and anti-IL-6 treatment alleviated bone loss. In vitro experiments showed that in the MC3T3-E1 osteoblast cell line exposed to microgravity, anti-IL-6 treatment significantly increased the transcriptional levels of *ALP*, *Runx2*, and *OPN* [34]. Further mechanistic studies found that IL-6 and its soluble receptor significantly reduced the expression of the osteoblast-related genes *ALP*, *Runx2*, *osterix*, and osteocalcin (*OCN*) in osteoblasts in a dose-dependent manner, and that IL-6 can inhibit osteoblast differentiation by activating the SHP2/PI3K/Akt2 and SHP2/MEK2/ERK pathways [42]. Moreover, IL-6 can activate the transcription factor STAT3, thereby promoting the binding of IGFBP5 to IGF-1 to inhibit osteoblast differentiation [24]. In addition, the induction of osteoarthritis can reduce the expression of the osteogenic marker serum type I collagen N-terminal propeptide. The preventive administration of an anti-IL-6R antibody can alleviate this condition, indicating that the inhibitory influence of IL-6 on osteoblast differentiation and maturation spans both physiological homeostasis and pathological states, as exemplified by its role in osteoarthritis progression [43]. Li et al. showed that IL-6 stimulation can promote the expression of *Runx2* and *OCN* in osteoblasts, but it can inhibit cell proliferation and induce apoptosis during the later stages of osteogenic differentiation [44].

A body of evidence indicates that IL-6 supports osteoblast differentiation. In the MG-63 human osteosarcoma cell line, the IL-6-sIL-6R complex promotes parathyroid hormone-related protein (PTHrP) production, which in turn stimulates osteoblast differentiation [25]. IL-6 can bind to mIL-6R to promote the phosphorylation of STAT3, promoting the osteogenic differentiation of BMSCs through a feedback loop of autocrine/paracrine [45]. In addition to acting on osteoblasts, IL-6 also mediates the transformation of vascular smooth muscle cells into osteoblasts. The IL-6/sIL-6R complex stimulates the histone demethylase Jumonji domain-containing protein 2B (JMJD2B) at the Runx2 promoter region stat binding site, which in turn demethylates histone H3 trimethyl K9 (histone 3 lysine 9, H3K9me3) to upregulate Runx2 expression and promote osteogenic differentiation [46].

## 3. Effects of Other Muscle-Derived Factors on Bone Homeostasis

In muscle–skeletal crosstalk, in addition to IL-6, there are many other muscle-derived factors, such as irisin, which positively regulates bone formation, insulin-like growth factor 1 (IGF-1), and myostatin, which negatively regulates bone formation [47].

### 3.1. Irisin

Irisin is produced by the cleavage of its precursor fibronectin type III domain 5 (FNDC5), which is produced by skeletal muscle and released into the circulation during exercise [48]. Bones are the primary target organ for iridoside, and recombinant irisin can enhance bone mass and mechanical properties in the cortical bone of young mice [49]. Irisin promotes the differentiation and activity of osteogenic lineage cells. Recombinant irisin treatment activates extracellular signal-regulated kinase (Erk) signal transduction, thereby upregulating the expression of osteogenesis-related genes such as *ATF4* and *Runx2* [49]. Irisin specifically interacts with the membrane-bound αV integrin receptor, inducing ERK/STAT3 cascade phosphorylation of BMP2/Smad signaling, thereby promoting the osteogenic differentiation of BMSCs [50]. In addition, irisin can competitively bind to TGF-β type II receptors, thereby weakening the inhibitory effect of TGFβ1 on Runx2 [51]. Irisin can prevent osteoclast activation by activating P38 and JNK [52], and regulates the balance between osteoblast-mediated bone formation and osteoclast-mediated bone resorption by inhibiting oxidative stress and RANKL production [53]. In addition, irisin has been shown to interact with IL-6, thereby regulating bone metabolism. Treatment with recombinant irisin suppresses the TLR4/MyD88/NF-κB signaling cascade, thereby reducing IL-6 secretion and restoring the ability of BMSCs to differentiate into osteoblasts [54].

### 3.2. IGF-1

As a member of the insulin-like growth factor family, IGF-1 exhibits a molecular architecture closely resembling that of insulin, particularly in conserved domain regions. It is mainly produced by the liver under the control of growth hormone and has an anabolic effect in adults [55]. Several splice variants of IGF-1 have been identified, among which IGF-1Ea and IGF-1Eb/c (MGF) play a role in muscle and skeletal homeostasis [56]. The 70 kD IGF-1 domain encoded by exons 3 and 4 of the IGF-1 gene promotes osteoblast survival, stimulates their differentiation, and promotes matrix production [57,58]. The 70 kD domain of IGF-1 has been shown to promote osteoclast formation [59].

### 3.3. Myostatin

Designated as growth and differentiation factor 8 (GDF-8), myostatin is grouped into the TGF-β superfamily and has been implicated in the repression of skeletal muscle growth and bone remodeling [60]. Myostatin promotes bone resorption by promoting osteoclast activity and inhibiting the bone formation function of osteoblasts, thereby hindering bone development [61]. BMSCs from myostatin knockout mice exhibit increased potential for osteogenic differentiation, while recombinant myostatin treatment inhibits the expression of *BMP2* and *IGF-1* in BMSCs, thereby suppressing their osteogenic differentiation capacity [62]. In addition, in the RA model of TNF-α transgenic mice, the high expression of myostatin in synovial tissue promotes *NFATC1* expression, thereby significantly promoting RANKL-mediated osteoclast differentiation [63].

## 4. Exercise-Mediated IL-6 Pathway Regulates Bone Metabolism

### 4.1. Effects of Exercise on IL-6 in the Skeletal Muscle

Cumulative evidence from prior investigations indicates that skeletal muscle possesses the capacity to synthesize and secrete cytokines originating from multiple molecular families. The first “myokine” identified to be released by muscle fiber contraction was IL-6. In the static state, the expression of IL-6 in skeletal muscle is low, and a small amount of IL-6 protein is present in type I fibers. However, the expression of IL-6 in skeletal muscle increases after 30 min of exercise. At the end of exercise, the expression of IL-6 further significantly increases, and the expression of IL-6 in blood circulation increases by 100 times [7]. Steensberg et al. also found that the net release of IL-6 in muscles during the last 2 h of exercise was 17 times higher than the increase in arterial IL-6 concentration, and that 5 h after exercise, the net release per minute was half the amount of IL-6 in the plasma. This notion indicates that the turnover rate of IL-6 during muscle exercise is high, and IL-6 produced by skeletal muscle may help maintain glucose homeostasis during prolonged exercise [4]. Further studies found that IL-6 skeletal muscle-specific knockout mice had significantly lower blood circulation concentrations of IL-6 after 50 min of exhausting exercise on a treadmill compared to wild-type mice, indicating that the vast majority of IL-6 molecules that increase in circulation during exercise come from skeletal muscle [64].

In addition, the increase in circulating IL-6 induced by exercise does not linearly increase over time but rather accelerates in an almost exponential manner. The peak of IL-6 concentrations is reached at the end of exercise or shortly afterward; thereafter, the liver quickly removes excess IL-6 from circulation, limiting the negative metabolic effects of a sustained increase in circulating IL-6 levels [65]. The release of IL-6 by muscles during exercise is closely related to the intensity and duration of exercise and is not closely related to the exercise mode. In knee-dominant exercises, such as running and cycling, a 10-fold increase in IL-6 in circulation requires 1.9 h of exercise. Meanwhile, a 100-fold increase in IL-6 in circulation requires 6.0 h of exercise. Therefore, only when the exercise involves a considerable amount of muscle mass and works for a considerable period will there be a significant increase in IL-6 in circulation. Otherwise, the increase in systemic IL-6 may be minimal or absent, making it difficult to achieve the expected systemic effect of IL-6 [66].

In the discussion above, we have learned that the effect of IL-6 on bone is multifaceted. However, considering that most of the IL-6 in circulation under exercise conditions comes from skeletal muscle, we will further explore the specific mechanism by which the body regulates bone metabolism under exercise conditions.

### 4.2. IL-6-OCN Cycle in the Musculoskeletal System Mediated by Exercise

The exercise capacity of mice is severely diminished when IL-6 is deficient in skeletal muscle. Blocking IL-6 with an IL-6 antibody has a similar effect, while exogenous OCN improves the exercise capacity of mice. The increase in OCN after exercise is closely related to the IL-6 of skeletal muscle origin, as the IL-6 OCN levels in the skeletal muscle conditional knockout mice do not change, while the OCN levels in control mice normally increase [31]. Further studies have found that myogenic IL-6 can bind to IL-6R on osteoblasts, leading to the phosphorylation of JAK, signal transducers, and activators of transcription (STAT), which triggers the increased expression of bone formation markers such as OCN and ALP [67]. OCN, which is derived from bone tissue, can interact with the G protein-coupled receptor C group 6 member A receptor in muscle fibers, resulting in the enhanced expression of the muscle factor IL-6. The upregulation of IL-6 further promotes the secretion of RANKL by osteoblasts and stimulates the differentiation of osteoclasts and bone resorption. The enhanced bone resorption process results in an increase in the release of OCN from the bone matrix. On the one hand, this virtuous cycle promotes the skeletal muscle’s adaptation to exercise by inducing catabolic metabolism, on the other hand, this cycle improves bone metabolism during exercise by increasing the operating efficiency of bone formation and resorption [64]. However, IL-6 in circulating serum binds directly to free IL-6Rα compared with myokine-derived IL-6, and the activation of sIL-6R inhibits osteogenic differentiation via the ERK or PI3K/AKT2 pathways, which is mediated by the SRC homology region 2 containing protein-tyrosine phosphatase (SHP2) [42,68]. The activation of SIL-6R in osteoclasts promotes the production of prostaglandin E2, which activates and promotes the differentiation of osteoclasts and their bone resorption function [69]. Accordingly, the coexistence of IL-6 and IL-6R in circulating serum is a direct cause of bone mass loss. This notion shows that the skeletal muscle-dominated IL-6-OCN cycle plays a key role in the body’s exercise metabolic balance. IL-6, a mediator or signaling factor, is secreted in increased amounts under the influence of exercise, thereby connecting muscle cells, osteoblasts, and osteoclasts to ensure muscle mobility and bone metabolic balance during exercise (Figure 2). Determining how to use the IL-6-OCN cycle in the musculoskeletal system to treat clinically relevant musculoskeletal disorders may become the focus of future research.

### 4.3. Exercise-Mediated IL-6 Promotes Fatty Acid Oxidation

The discovery of muscle factors, fat factors, and bone-derived factors that can act on nearby organs through endocrine/paracrine mechanisms has provided a new perspective for understanding the muscle–fat–bone trilogy [70,71]. Exercise can cause a variety of metabolic and physiological changes in the body. Research on the regulation of bone metabolism by exercise is currently focused on signal pathways, cytokines, and non-coding RNAs [72,73,74]. Osteoblasts are essential for bone growth and maintenance, and clinical diseases of substrate utilization (such as diabetes, obesity, and aging) can lead to osteoblast dysfunction, leading to bone fragility and osteoporotic fractures [75]. Osteoblasts need oxidative fatty acid energy during differentiation, accounting for about 40–80% of glucose energy production [76]. Fatty acid oxidation significantly increases as osteoblasts mature in vitro, and their catabolic activity level is three times that of proliferating cells. In vitro, the pharmacological inhibition of β-oxidation can impair osteoblast differentiation [77]. During the late stage of differentiation, osteoblasts induce the expression of the key enzymes required for fatty acid oxidation by expressing the Wnt co-receptor Lrp5 due to the production of matrix and the need for a large amount of energy for bone mineralization. Furthermore, the rough endoplasmic reticulum helps collect energy to complete this process [78,79]. During the later stages of differentiation, osteoblasts induce the expression of key enzymes required for fatty acid oxidation by expressing the Wnt co-receptor Lrp5 due to the production of matrix and the need for a large amount of energy for bone mineralization. Moreover, the rough endoplasmic reticulum helps collect energy. In several in vitro and in vivo studies, IL-6 injection therapy has been found to increase local or systemic fatty acid oxidation, including osteoblasts. After the second hour of human recombinant IL-6 injection, the rate of fatty acid oxidation significantly increased. The main mechanism for this is the content of AMP-activated protein kinase (AMPK) increasing after injection. AMPK can phosphorylate acetyl-CoA carboxylase (ACC), thereby inhibiting ACC activity, resulting in a decrease in the content of malonyl-CoA, relieving the inhibitory effect on CPT-1 and increasing the oxidation of fatty acids to complete the process [80,81,82]. The complete oxidation of fatty acids can produce three times as much ATP compared to glucose molecules [83]. Therefore, exercise can promote the osteogenic differentiation process by increasing the level of the muscle factor IL-6 in circulation and increasing the oxidation of fatty acids in or around osteoblasts.

### 4.4. IL-6 Pro-Inflammatory Effect Mediated by Exercise

In addition to the role of IL-6 as an inflammatory factor, as a myokine, it is an important pathogenic factor associated with immune-mediated bone diseases. In addition to the benefits of exercise-induced IL-6, such as promoting OCN secretion and fatty acid oxidation, it also exacerbates local and systemic osteoclast-mediated bone destruction, including Paget’s disease of bone (PDB), rheumatoid arthritis (RA), and the bone metastasis of breast cancer [84,85,86]. Bone loss and subsequent localized bone erosion are a prominent feature of the above diseases, which are mainly manifested as increased osteoclast activity [87]. IL-6 can act on osteoblasts to upregulate RANKL expression and promote binding to its ligand RANK, promoting osteoclast differentiation and aggravating the above diseases. PDB is a common bone metabolic disorder disease, the main cause of which is increased bone remodeling and an abnormal bone structure caused by abnormal osteoclasts. The IL-6 content in the serum of PDB patients is significantly increased. IL-6 can increase the RANKL sensitivity of osteoclast precursors, increase the number of osteoclasts, and aggravate the development of PDB [88]. The abnormally high concentration of IL-6 in RA patients is considered to be a key factor leading to increased bone resorption in these subjects [89]. Tocilizumab (TCZ) is a humanized anti-IL-6R antibody recommended for the clinical treatment of RA. This antibody blocks the IL-6 regulatory pathway by binding to mIL-6R and sIL-6R and inhibits the pro-inflammatory effect of IL-6 by inhibiting the dimerization of gp130 [90]. TCZ, in combination with methotrexate (MTX), significantly reduces bone resorption markers. In addition, TCZ and MTX combination therapy significantly improves lumbar spine and femoral neck bone mineral density in RA patients with osteoporosis [91]. In mice with an arthritis model, IL-6 specifically regulates the local production of T lymphocyte IL-17, RANKL, and OPG during inflammatory osteoarthritis. The absence of IL-6 expression, as observed in *IL-6*^−/−^ mice, leads to impaired osteoclast localization within inflamed joint tissues, the antigen-specific proliferation responses of CD4+ T lymphocytes, and RANKL/OPG [86]. Blocking IL-6 for the treatment of RA may have a particularly beneficial effect on T lymphocyte-mediated osteoclast recruitment and activation. Consistent with these observations, recent studies have shown that treatment with the IL-6R inhibitor tocilizumab can inhibit biochemical markers of osteoclast-mediated bone destruction in RA patients [92]. Accordingly, understanding the disadvantages of myokine factors in sports for the above-mentioned diseases can improve the formulation of exercise prescriptions for patients in the clinic. IL-6 has been implicated as a critical mediator in oncogenesis, in the spread of malignant cells to distant sites, and in the maintenance of a pro-tumorigenic inflammatory microenvironment, as a pleiotropic factor of the myokine inflammatory factor [93,94]. Elucidating the mechanistic basis by which IL-6’s pleiotropic effects govern disease trajectory—encompassing progression kinetics, severity gradients, and temporal dynamics—alongside delineating the molecular architecture underpinning IL-6’s pathogenic involvement across diverse disorders, will ultimately advance our understanding of canonical IL-6 receptor signaling pathways and trans-signaling mechanisms in autoimmune pathogenesis, infectious disease progression, epithelial barrier integrity, and tumorigenesis. This conceptual framework is anticipated to yield novel therapeutic insights and enhance clinical response rates.

## 5. Conclusions

In this review, we first describe the complex regulatory role of IL-6 in bone metabolism. IL-6 mainly promotes the maturation of osteoclasts by indirectly inducing the secretion of RANKL in osteoblasts, BMSCs, and osteocytes. In addition, IL-6 can regulate osteoclast maturation via the trans-signaling pathway in opposite ways, depending on the level of RANKL. Under physiological conditions, IL-6 does not appear to have a unique destructive effect on bone. However, in pathological states, IL-6 is the key to the excessive mediation of osteoclast bone resorption function to disrupt the balance of bone metabolism. In osteoblast cells, IL-6 plays a double-edged role. However, little research has been conducted on this aspect. Finally, we summarize several possible mechanisms by which IL-6 secreted by skeletal muscle regulates bone metabolism under exercise conditions, including exercise-mediated myoskeletal IL-6-OCN circulation, IL-6-promoted fatty acid oxidation, and exacerbated IL-6 inflammatory effects.

This review aims to elucidate the regulatory mechanisms by which physical exercise modulates musculoskeletal metabolic processes, providing a scientific rationale for determining the appropriate frequency, intensity, duration, and type of exercise; reasonably formulating personalized exercise prescriptions; and applying exercise therapy to treat and prevent musculoskeletal system diseases. However, factors such as genetics, age, gender, overall health status, and individual responses to exercise may result in significant variations in IL-6 secretion. Furthermore, while physical activity has positive effects on bone health, other important factors influencing bone metabolism, such as nutritional intake, psychological stressors, genetic susceptibility, and chronic diseases, must also be considered. Therefore, future research should incorporate these factors into studies examining IL-6-mediated muscle–bone crosstalk under exercise conditions to comprehensively elucidate the underlying mechanisms. In addition, other muscle factors will be discovered with the continuous development of multi-omics technology. The discovery of new muscle factors and their receptors may become a new research topic for treating musculoskeletal diseases.

## Figures and Tables

**Figure 1 biomolecules-15-00893-f001:**
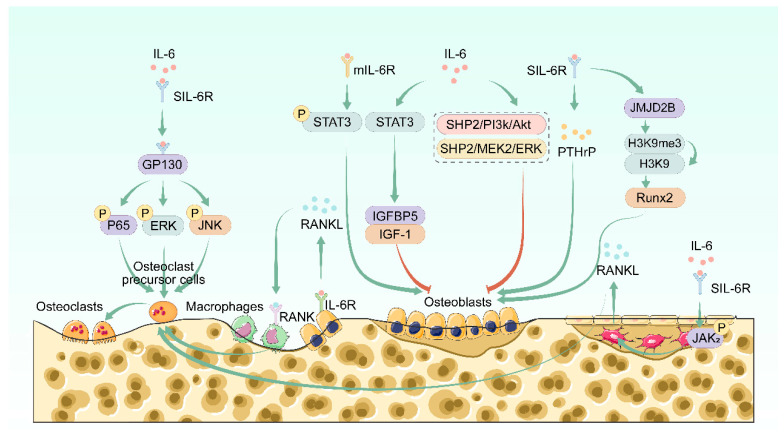
Effects of IL-6 on bone homeostasis. IL-6 plays a bidirectional regulatory role in the balance between osteoblast-mediated bone formation and osteoclast-mediated bone resorption. IL-6 can promote the secretion of RANKL by osteoblasts and osteocytes by binding to receptors on the surface of osteoblasts and osteocytes, thereby inducing the differentiation of osteoclasts. The trans-signaling mediated by the IL-6-sIL-6R complex can regulate the differentiation ability of RANKL-induced osteoclast precursors. In osteoblasts, IL-6 can exert a bidirectional regulatory function in osteogenic differentiation through different pathways.

**Figure 2 biomolecules-15-00893-f002:**
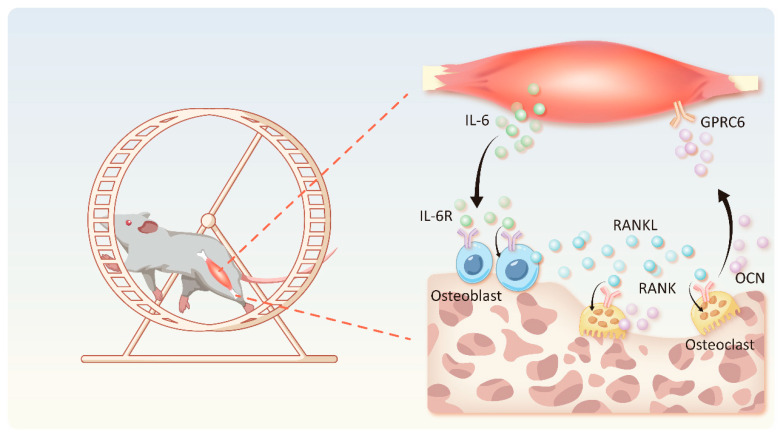
The IL-6-OCN cycle in the musculoskeletal system is mediated by exercise. The skeletal muscle-dominated IL-6-OCN cycle plays a key role in the metabolic balance of the myoskeletal system during exercise. The secretion of IL-6 in the skeletal muscle is increased under the influence of exercise. After binding to IL-6R on the surface of osteoblasts, IL-6-OCN promotes the secretion of RANKL, which induces the differentiation of osteoclasts, promotes the secretion of OCN by osteoclasts, and binds to the GPRC6 receptor in skeletal muscle, resulting in the enhanced expression of the myokine IL-6.

## Data Availability

No new data were created or analyzed in this study. Data sharing is not applicable to this article.

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
