# Peer review of "Exercise-Mediated Skeletal Muscle-Derived IL-6 Regulates Bone Metabolism: A New Perspective on Muscle–Bone Crosstalk"

_biomolecules, 2025, doi:10.3390/biom15060893_

Round 1

Reviewer 1 Report

Comments and Suggestions for Authors

This review highlights the pivotal role of IL-6 in bone metabolism, particularly its effects on osteoclast and osteoblast cells. According to the authors, the current literature suggests that IL-6 promotes osteoclast maturation by indirectly inducing RANKL secretion in osteoblasts, bone marrow stromal cells (BMSCs), and osteocytes. Notably, under physiological conditions, IL-6 does not seem to exert a unique destructive effect on bone. However, in pathological states, IL-6 plays a crucial role in excessive osteoclast-mediated bone resorption, disrupting the delicate balance of bone metabolism.

The authors also elucidate several potential mechanisms by which skeletal muscle-derived IL-6 regulates bone metabolism under exercise conditions. These mechanisms include exercise-mediated myosteal IL-6-OCN circulation, IL-6-promoted fatty acid oxidation, and exacerbated IL-6 inflammatory effects.

Overall, the review is well-written and effectively underscores the role of IL-6 in bone metabolism, particularly in the context of exercise. The authors provide a comprehensive summary of the latest data on this topic.

To further enhance the review, it would be beneficial to address the following queries:

  1. Does this review offer a new perspective on muscle-bone crosstalk, or does it build upon existing knowledge?
  2. How does the level of RANKL influence IL-6 role in osteoclast maturation via the trans-signaling pathway, please add few lines.
  3. Please add a paragraph on the role of  other muscle-secreted factors, besides IL-6, which are involved in regulating bone metabolism, and how do they interact with IL-6.
  4. There are many typographical mistakes that require attention to be addressed.

Elaborating on these points would provide a more understanding of IL-6 role in bone metabolism and its interactions with other factors.

Author Response

Dear Editor:

        Thank you very much to the editor and review experts for reviewing our manuscript in your busy schedules, your professional opinions are very important for the revision of our manuscript. We have revised the original text according to the comments of the reviewing experts and the revised content we have marked in blue. The following is our response to the reviewer's comments.

Reviewer 1:

Comment 1: Does this review offer a new perspective on muscle-bone crosstalk, or does it build upon existing knowledge?

        Reply: This review provides a new perspective on muscle-skeletal crosstalk, with mechanisms such as the regulatory role of IL-6 playing a significant role in bone metabolic diseases. Exercise, as an economical, convenient, and low-side-effect treatment method, plays a crucial role in improving musculoskeletal diseases. However, there is currently limited summary of how exercise participates in the regulation of bone metabolism through skeletal muscle-derived IL-6. Therefore, this paper takes skeletal muscle-derived IL-6 as the entry point, summarizes existing relevant literature, and explores how muscle-targeted cross-organ regulatory activities during exercise influence bone metabolic processes. This aims to refine the underlying mechanisms of muscle-bone crosstalk under exercise conditions, providing multi-faceted theoretical foundations and clinical diagnostic and therapeutic strategies for exercise-induced improvements in bone health.

Comment 2: How does the level of RANKL influence IL-6 role in osteoclast maturation via the trans-signaling pathway, please add few lines.

Reply: We added a discussion of the possible mechanism by which IL-6 trans-signaling regulates osteoclast maturation at different RANKL levels to the original text.

Here is the content with our additions (lines 1160-177 in the original text):

Further mechanistic studies have found that the IL-6-sIL-6R complex specifically en-hances and inhibits the phosphorylation of the NF-κB subunit p65, ERK, and JNK pathways in the presence of low and high levels of RANKL, thereby mediating osteo-clast maturation. The RANKL-dependent regulation of osteoclast maturation by IL-6 trans-signaling may reflect pathological processes in the local microenvironment asso-ciated with autoimmune diseases related to bone destruction, such as rheumatoid ar-thritis. Whether the effect is stimulatory or inhibitory may depend on the disease stage. In the early stages of the disease, when RANKL levels in the local microenvironment are relatively low, high levels of IL-6 and sIL-6R secreted by active immune cells in-crease osteoclast differentiation and function in two ways: (1) indirectly stimulating matrix cells or osteoblasts to produce RANKL, and (2) directly promoting the commit-ment of osteoclast precursors to mature osteoclasts. Both mechanisms activate osteo-clast-mediated bone resorption and lead to bone and joint cartilage erosion. However, in the progressive or late stages of the disease, when RANKL levels are high, IL-6 and sIL-6R may contribute to balancing the high RANKL concentrations produced in the bone microenvironment and inhibiting RANKL-induced osteoclast maturation, thereby protecting bones from excessive resorption. In this case, the IL-6-sIL-6R complex re-flects a protective mechanism on bones [40].

Comment 3: Please add a paragraph on the role of  other muscle-secreted factors, besides IL-6, which are involved in regulating bone metabolism, and how do they interact with IL-6.

        Reply: We have added the following paragraph to the original text as per your suggestion: “3. Effects of other muscle-derived factors on bone homeostasis.”This section explains the regulatory roles of other representative muscle factors in bone metabolism. Additionally, based on existing literature, we have included how other muscle factors interact with IL-6 to regulate bone metabolism; however, related studies are relatively scarce.

Here is the content with our additions (lines 1160-177 in the original text):

  1. Effects of other muscle-derived factors on bone homeostasis

        In muscle-skeletal crosstalk, in addition to IL-6, there are many other mus-cle-derived factors such as irisin, which positively regulates bone formation, insu-lin-like growth factor 1 (IGF-1), and myostatin, which negatively regulates bone for-mation [49].

3.1 Irisin

        Irisin is produced by the cleavage of its precursor fibronectin type III domain 5 (FNDC5), which is produced by skeletal muscle and released into the circulation during exercise [50]. Bones are the primary target organ for iridoside, and recombinant irisin can enhance bone mass and mechanical properties in the cortical bone of young mice [51]. Irisin promotes the differentiation and activity of osteogenic lineage cells. Recom-binant irisin treatment activates extracellular signal-regulated kinase (Erk) signal transduction, thereby upregulating the expression of osteogenesis-related genes such as ATF4 and Runx2 [51]. Irisin binds to the membrane-anchored receptor αV integrin, in-ducing ERK/STAT3 cascade phosphorylation of BMP2/Smad signaling, thereby pro-moting osteogenic differentiation of BMSCs [52]. In addition, irisin can competitively bind to TGF-β type II receptors, thereby weakening the inhibitory effect of TGFβ1 on Runx2 [53]. Irisin can prevent osteoclast activation by activating P38 and JNK [54], and regulates the balance between osteoblast-mediated bone formation and osteo-clast-mediated bone resorption by inhibiting oxidative stress and RANKL production [55]. In addition, irisin has been shown to interact with IL-6, thereby regulating bone metabolism. The application of recombinant irisin can inhibit the activation of the TLR4/MyD88/NF-κB pathway, thereby reducing IL-6 secretion and restoring the ability of BMSCs to differentiate into osteoblasts [56].

3.2 IGF-1

IGF-1 is a hormone with a molecular structure very similar to insulin. It is mainly produced by the liver under the control of growth hormone and has an anabolic effect in adults [57]. Several splice variants of IGF-1 have been identified, among which IGF-1Ea and IGF-1Eb/c (MGF) play a role in muscle and skeletal homeostasis [58]. The 70 kD IGF-1 domain encoded by exons 3 and 4 of the IGF-1 gene promotes osteoblast survival, stimulates their differentiation, and promotes matrix production [59, 60]. The 70 kD domain of IGF-1 has been shown to promote osteoclast formation [61].

3.3 Myostatin

Myostatin, also known as growth and differentiation factor 8 (GDF-8), belongs to the transforming growth factor β (TGF-β) superfamily and is considered a negative regulator of skeletal muscle and bone [62]. Myostatin promotes bone resorption by promoting osteoclast activity and inhibiting the bone formation function of osteoblasts, thereby hindering bone development and bone injury repair [63]. BMSCs from myo-statin knockout mice exhibit enhanced osteogenic differentiation capacity, while re-combinant myostatin treatment inhibits the expression of BMP2 and IGF-1 in BMSCs, thereby suppressing their osteogenic differentiation capacity [64]. In addition, in the RA model of TNF-α transgenic mice, high expression of myostatin in synovial tissue promotes NFATC1 expression, thereby significantly promoting RANKL-mediated os-teoclast differentiation [65].

Comment 4: There are many typographical mistakes that require attention to be addressed.

Reply: We have corrected the typographical errors in the text.

These are the changes we made to the entire manuscript based on the reviewer's comments. Thanks again for taking the time to review this manuscript. We appreciate all your generous comments and suggestions!

Yours Sincerely!

Reviewer 2 Report

Comments and Suggestions for Authors

1
Title
The Title of the manuscript is consistent with its content. However, please include that this is a narrative review.
Abstract
The Abstract provides an explicit statement of the main objectives of the study. There are only minor areas for improvement:
1) line 23: if “few”, then “summaries” instead of “summary”. Actually, I would write: “The exact mechanism by which IL-6 regulates bone metabolism is not fully understood, and there are few summaries on how exercise affects bone metabolism through IL-6 from skeletal muscles."
2) line 27: not “aims improve” but “aims to improve”.
3) line 27/28: please add “a” before “theoretical”.
1. Introduction
4) The Introduction effectively describes the rationale for the review within the context of existing knowledge.
5) The Introduction indicates the primary aim of investigating the effect of exercise-mediated muscle-bone "crosstalk" on bone metabolism, particularly through the lens of the muscle factor IL-6.
6) lines 47-48: “However, the skeletal muscle can also affect bone homeostasis in non-mechanical ways [5].“ Please replace this with the following: “However, the skeletal muscle can also non-mechanically affect bone homeostasis [5].”
2. Effects of IL-6 on bone homeostasis
2.1. IL-6 structure and function
7) Please add references after “IL-6 is a pleiotropic cytokine. “ This is in line 63.
8) I noticed that very often “levels are” used. If concentrations were meant in sentences, then replace “level” with “concentration”.
9) lines 66-67: please change this sentence: “Moreover, IL-6 is considered to be one of the main immunomodulatory cytokines that control health and disease[9]”. Write “Moreover, IL-6 is considered one of the main immunomodulatory cytokines controlling health and disease [9].”
2.2. IL-6 regulates bone homeostasis
10) lines 89-90: please replace “Bone homeostasis is essential for the formation and maintenance of bone form and the repair of damaged bone[19].” with the following: “Bone homeostasis is essential for forming and maintaining bone form and repairing damaged bone [19].”
11)lines 101- 107: Figure 1. Usually, figure titles are below the figures.
2
2.2.1. IL-6 regulates bone resorption
12) lines 117-119: “However, IL-6 has no significant effect on the differentiation of purified osteoclast precursor cells [27-30].” Please replace with “However, IL-6 does not significantly affect the differentiation of purified osteoclast precursor cells [27-30]. “
13) line 141: please replace “does not have a “ with “has no”.
14) lines 151-152: “this effect can be inhibited by anti-gp130 antibodies but not by osteoprotegerin (OPG), indicating that IL-6 can directly induce osteoclast differentiation through a RANKL-independent mechanism[38].” This should be a new sentence starting with a capital letter, so no semicolon before this sentence.
15) lines 162-163: “This nition indicates that IL-6 plays a bidirectional regulatory 162 role in RANKL-induced osteoclastogenesis in the presence of sIL-6R. “ What is “nition”?
Latin phrases such as “in vitro” should be written in italics. Line 186.
2.2.2. IL-6 regulates bone formation
16) line 186: “In vitro, experiments”-please remove the comma
3. Exercise-mediated IL-6 pathway regulates bone metabolism
3.1. Effects of exercise on IL-6 in the skeletal muscle
17) I noticed that very often “levels are” used. For example: “blood circulation levels of IL-6 after 50 min” in line 227. The word “levels” should be replaced with “ concentrations”. Another example: “The peak of IL-6 levels is reached at the end of exercise or shortly afterward; thereafter, the liver quickly removes excess IL-6 from circulation, limiting the negative metabolic effects of a sustained increase in circulating IL-6 levels [50]. “ in lines 232-235. Please revise entire manuscript and if concentrations are meant in sentences then replace “level” with “concentration”.
18) line 225: please replace “in maintaining” with “maintain”.
3.2. IL-6-OCN cycle in the musculoskeletal system mediated by exercise
19) I noticed that often “levels are” used. If concentrations were meant in sentences, then replace “level” with “concentration”.
20) lines 278- 284: Figure 2. Usually, figure titles are below the figures.
3.3. Exercise-mediated IL-6 promotes fatty acid oxidation
21) Lines 301 and 305: please replace “in collecting “ with “collect”.
22) Latin phrases such as “in vivo” should be written in italics. Line 306.
3
23) Latin phrases such as “in vitro” should be written in italics. Lines 296, 306.
3.4. IL-6 pro-inflammatory effect mediated by exercise
24) lines 336-337: TCZ in combination with methotrexate (MTX) significantly reduced bone resorption markers. Please add commas, it should be: “TCZ, in combination with methotrexate (MTX), significantly reduced bone resorption markers.”
4. Conclusion
25) line 381: please replace “the treatment of” with “treating”.
26) Conclusions are shorter sections of academic texts that usually serve two functions. The first is to summarise and bring together the main areas covered in writing, which might be called ‘looking back’, and the second is to give a final comment or judgment. This section could benefit from improvements in structure, clarity, and conciseness.
27) In general, bone metabolism is influenced by multiple systemic factors, and isolating the impact of skeletal muscle-derived IL-6 from other pathways or influences can be challenging. It is understood that the Authors tried to isolate the IL-6, which is fine. However, the response to exercise and the resulting IL-6 secretion can vary widely among individuals due to genetics, age, sex, and overall health, which may complicate the interpretation of findings. While physical activity positively influences bone health, the study may not comprehensively consider other critical lifestyle factors, such as nutritional intake, psychological stressors, and genetic predispositions, which play significant roles in bone metabolism. Additionally, the results may be more relevant to specific demographic groups, such as young, healthy individuals. They may not directly apply to other populations, including the elderly or those with chronic health conditions.
General remarks:
28) The increasing attention towards myokines is noteworthy. The characterization of IL-6 as a bidirectional modulator of bone homeostasis contributes significant insights to this area of research.
29) Please add a space before references. I noticed there is no space. Revise the entire manuscript. E.g., “The frequent use of electronic devices, sedentary lifestyles, and a high-fat, high-sugar diet, driven by the continuous advancement of urbanization and industrialization around the world, pose a significant threat to public health[1].” This is the first sentence in lines 33-35, and this is just one example.
30) Line 392: “Acknowledgments: We appreciate the time and effort of the participants in this study. “ This sounds odd. The manuscript is a review and did not enroll participants. So it looks like the Authors thank themselves. I would remove this sentence, write “Not applicable”, or acknowledge somebody.
31) The manuscript has substantial scientific, practical, and educational value.

Comments on the Quality of English Language

The English could be improved to more clearly express the research. The writing is mostly clear, with a logical flow of ideas.  However, some sentences could be simplified for better readability without sacrificing content. For example, instead of: "Moreover, IL-6, as a multi-effect factor, has a complex regulatory effect on bone metabolism." in lines 54-55, the following could be used: "IL-6 is a multi-functional factor that has a complex role in regulating bone metabolism."

Author Response

Dear Editor:

        Thank you very much to the editor and review experts for reviewing our manuscript in your busy schedules, your professional opinions are very important for the revision of our manuscript. We have revised the original text according to the comments of the reviewing experts and the revised content we have marked in blue. The following is our response to the reviewer's comments.

Reviewer 2:

Comment 1:

line 23: if “few”, then “summaries” instead of “summary”. Actually, I would write:“The exact mechanism by which IL-6 regulates bone metabolism is not fully understood, and there are few summaries on how exercise affects bone metabolism through IL-6 from skeletal muscles."

Reply: We have replaced the original sentence with the one you suggested. (line 22-24)

Comment 2:

line 27: not“aims improve”but“aims to improve”.

Reply: We have made the changes according to your suggestions. (line 26)

Comment 3:

        line 27/28: please add “a” before “theoretical”.

        Reply: We have added “a” before “theoretical”. (line 27)

Comment 4:

The Introduction effectively describes the rationale for the review within the context of existing knowledge.

Reply: Thank you.

Comment 5:

The Introduction indicates the primary aim of investigating the effect of exercise-mediated muscle-bone "crosstalk" on bone metabolism, particularly through the lens of the muscle factor IL-6.

Reply: Thank you.

Comment 6:

        lines 47-48: “However, the skeletal muscle can also affect bone homeostasis in non-mechanical ways [5].“ Please replace this with the following: “However, the skeletal muscle can also non-mechanically affect bone homeostasis [5].”

        Reply: We have made the changes according to your suggestions. (line 46-47)

Comment 7:

        Please add references after“IL-6 is a pleiotropic cytokine.” This is in line 63.

        Reply: We have added references after “IL-6 is a pleiotropic cytokine.” (line 61)

Comment 8:

        I noticed that very often “levels are” used. If concentrations were meant in sentences, then replace “level” with “concentration”.

        Reply: The “levels” in the review does not mean “concentrations”.

Comment 9:

        lines 66-67: please change this sentence: “Moreover, IL-6 is considered to be one of the main immunomodulatory cytokines that control health and disease[9]”. Write “Moreover, IL-6 is considered one of the main immunomodulatory cytokines controlling health and disease [9].”

Reply: We have made the changes according to your suggestions. (line 64-65)

Comment 10:

lines 89-90: please replace“Bone homeostasis is essential for the formation and maintenance of bone form and the repair of damaged bone[19].”with the following: “Bone homeostasis is essential for forming and maintaining bone form and repairing damaged bone [19].”

        Reply: We have made the changes according to your suggestions. (line 87-88)

Comment 11:

lines 101- 107: Figure 1. Usually, figure titles are below the figures.  

        Reply: We have placed the figure title below Figure 1. (line 98-104)

Comment 12:

        lines 117-119: “However, IL-6 has no significant effect on the differentiation of purified osteoclast precursor cells [27-30].” Please replace with “However, IL-6 does not significantly affect the differentiation of purified osteoclast precursor cells [27-30].

Reply: We have made the changes according to your suggestions. (line 114-115)

Comment 13:

        line 141: please replace “does not have a” with “has no”.

Reply: We have replaced  “does not have a” with “has no”. (line 138)

Comment 14:

        lines 151-152: “this effect can be inhibited by anti-gp130 antibodies but not by osteoprotegerin (OPG), indicating that IL-6 can directly induce osteoclast differentiation through a RANKL-independent mechanism[38].” This should be a new sentence starting with a capital letter, so no semicolon before this sentence.

Reply: We have made the changes according to your suggestions. (line 147)

Comment 15:

        lines 162-163: “This nition indicates that IL-6 plays a bidirectional regulatory 162 role in RANKL-induced osteoclastogenesis in the presence of sIL-6R. “ What is “nition”? Latin phrases such as “in vitro” should be written in italics. Line 186.

        Reply: The “nition” is “ notion”, we have changed it. We have changed “in vitro”  to “In vitro”. (line 159)

Comment 16:

        line 186:“In vitro, experiments”-please remove the comma

        Reply: We have removed the comma. (line 197)

Comment 17:

        I noticed that very often “levels are”used. For example: “blood circulation levels of IL-6 after 50 min”in line 227. The word“levels”should be replaced with “ concentrations”. Another example: “The peak of IL-6 levels is reached at the end of exercise or shortly afterward; thereafter, the liver quickly removes excess IL-6 from circulation, limiting the negative metabolic effects of a sustained increase in circulating IL-6 levels [50]. “in lines 232-235. Please revise entire manuscript and if concentrations are meant in sentences then replace “level” with “concentration”.

Reply: We have replaced “level” with “concentration”. (line 281)

Comment 18:

line 225: please replace“in maintaining” with “maintain”.

        Reply: We have replaced “in maintaining” with “maintain”. (line 279)

Comment 19:

        I noticed that often “levels are” used. If concentrations were meant in sentences, then replace “level” with “concentration”.

        Reply: Here, The “levels” does not mean “concentrations”.

Comment 20:

        lines 278- 284: Figure 2. Usually, figure titles are below the figures.

Reply: We have placed the figure title below Figure 2. (line 332-338)

Comment 21:

Lines 301 and 305: please replace “in collecting “ with “collect”.

Reply: We have replaced  “in collecting”  with “collect”. (line 355, 359)

Comment 22:

Latin phrases such as “in vivo” should be written in italics. Line 306.

Reply: We have made the changes according to your suggestions. (line 360)

Comment 23:

Latin phrases such as “in vitro” should be written in italics. Lines 296, 306.

Reply: We have made the changes according to your suggestions. (line 360)

Comment 24:

        lines 336-337: TCZ in combination with methotrexate (MTX) significantly reduced bone resorption markers. Please add commas, it should be: “TCZ, in combination with methotrexate (MTX), significantly reduced bone resorption markers.”

Reply: We have made the changes according to your suggestions. (line 390-391)

Comment 25:

line 381: please replace “the treatment of” with “treating”.

Reply: We have replace “the treatment of” with “treating”. (line 437)

Comment 26:

        Conclusions are shorter sections of academic texts that usually serve two functions. The first is to summarise and bring together the main areas covered in writing, which might be called ‘looking back’, and the second is to give a final comment or judgment. This section could benefit from improvements in structure, clarity, and conciseness.

        Reply: We have optimized the structure, clarity, and conciseness of the conclusions according to your suggestions.

Here is the content with our additions (line 412-437) in the original text:

In this review, we first describe the complex regulatory role of IL-6 in bone metab-olism. IL-6 mainly promotes the maturation of osteoclasts by indirectly inducing the secretion of RANKL in osteoblasts, BMSCs, and osteocytes. In addition, IL-6 can regu-late osteoclast maturation via the trans-signaling pathway in opposite ways, depending on the level of RANKL. Under physiological conditions, IL-6 does not appear to have a unique destructive effect on bone. However, in pathological states, IL-6 is key to the excessive mediation of osteoclast bone resorption function to disrupt the balance of bone metabolism. In osteoblast cells, IL-6 plays a double-edged role. However, few re-search has been conducted on this aspect. Finally, we summarize several possible mechanisms by which IL-6 secreted by skeletal muscle regulates bone metabolism un-der exercise conditions, including exercise-mediated myosteal IL-6-OCN circulation, IL-6-promoted fatty acid oxidation, and exacerbated IL-6 inflammatory effects.

This study hopes to reveal that exercise can regulate the body’s musculoskeletal metabolism, providing a theoretical basis for scientifically selecting the frequency, in-tensity, duration, and type of exercise, reasonably formulating personalized exercise prescriptions, and applying exercise therapy to treat and prevent musculoskeletal sys-tem diseases. However, factors such as genetics, age, gender, overall health status, and individual responses to exercise may result in significant variations in IL-6 secretion. Furthermore, while physical activity has positive effects on bone health, other im-portant factors influencing bone metabolism, such as nutritional intake, psychological stressors, genetic susceptibility, and chronic diseases, must also be considered. There-fore, future research should incorporate these factors into studies examining IL-6-mediated muscle-bone crosstalk under exercise conditions to comprehensively elucidate the underlying mechanisms. In addition, other muscle factors will be discov-ered with the continuous development of multi-omics technology. The discovery of new muscle factors and their receptors may become a new research topic for treating musculoskeletal diseases.

Comment 27:

        In general, bone metabolism is influenced by multiple systemic factors, and isolating the impact of skeletal muscle-derived IL-6 from other pathways or influences can be challenging. It is understood that the Authors tried to isolate the IL-6, which is fine. However, the response to exercise and the resulting IL-6 secretion can vary widely among individuals due to genetics, age, sex, and overall health, which may complicate the interpretation of findings. While physical activity positively influences bone health, the study may not comprehensively consider other critical lifestyle factors, such as nutritional intake, psychological stressors, and genetic predispositions, which play significant roles in bone metabolism. Additionally, the results may be more relevant to specific demographic groups, such as young, healthy individuals. They may not directly apply to other populations, including the elderly or those with chronic health conditions.

        Reply: Based on your suggestion, we have included the influence of factors such as genetics, age, and gender on muscle IL6 secretion and the skeletal regulatory effects of physical activity in our conclusions and provided a brief discussion on this topic. (line 424-434)

Comment 28:

The increasing attention towards myokines is noteworthy. The characterization of IL-6 as a bidirectional modulator of bone homeostasis contributes significant insights to this area of research.

Reply: Thank you.

Comment 29:

        Please add a space before references. I noticed there is no space. Revise the entire manuscript. E.g., “The frequent use of electronic devices, sedentary lifestyles, and a high-fat, high-sugar diet, driven by the continuous advancement of urbanization and industrialization around the world, pose a significant threat to public health[1].” This is the first sentence in lines 33-35, and this is just one example.

Reply: We have made the changes according to your suggestions.

Comment 30:

        Line 392: “Acknowledgments: We appreciate the time and effort of the participants in this study. “ This sounds odd. The manuscript is a review and did not enroll participants. So it looks like the Authors thank themselves. I would remove this sentence, write “Not applicable”, or acknowledge somebody.

Reply: We have made the changes according to your suggestions. (line 448)

Comment 31:

        The manuscript has substantial scientific, practical, and educational value.

Reply: Thank you.

Comments on the Quality of English Language

        The English could be improved to more clearly express the research. The writing is mostly clear, with a logical flow of ideas.  However, some sentences could be simplified for better readability without sacrificing content. For example, instead of: "Moreover, IL-6, as a multi-effect factor, has a complex regulatory effect on bone metabolism." in lines 54-55, the following could be used: "IL-6 is a multi-functional factor that has a complex role in regulating bone metabolism."

Reply: We have made the changes according to your suggestions. (line 53-54)

These are the changes we made to the entire manuscript based on the reviewer's comments. Thanks again for taking the time to review this manuscript. We appreciate all your generous comments and suggestions!

Yours Sincerely!
